# Targeting Angiogenesis in Prostate Cancer

**DOI:** 10.3390/ijms20112676

**Published:** 2019-05-31

**Authors:** Zsombor Melegh, Sebastian Oltean

**Affiliations:** 1Department of Cellular Pathology, Southmead Hospital, Bristol BS10 5NB, UK; zsombor.melegh@nbt.nhs.uk; 2Institute of Biomedical and Clinical Sciences, Medical School, College of Medicine and Health, University of Exeter, Exeter EX12LU, UK

**Keywords:** prostate cancer, angiogenesis, VEGF-A, splicing isoforms

## Abstract

Prostate cancer is the most commonly diagnosed cancer among men in the Western world. Although localized disease can be effectively treated with established surgical and radiopharmaceutical treatments options, the prognosis of castration-resistant advanced prostate cancer is still disappointing. The objective of this study was to review the role of angiogenesis in prostate cancer and to investigate the effectiveness of anti-angiogenic therapies. A literature search of clinical trials testing the efficacy of anti-angiogenic therapy in prostate cancer was performed using Pubmed. Surrogate markers of angiogenic activity (microvessel density and vascular endothelial growth factor A (VEGF-A) expression) were found to be associated with tumor grade, metastasis, and prognosis. Six randomizedstudies were included in this review: two phase II trials on localized and hormone-sensitive disease (*n* = 60 and 99 patients) and four phase III trials on castration-resistant refractory disease (*n* = 873 to 1224 patients). Although the phase II trials showed improved relapse-free survival and stabilisation of the disease, the phase III trials found increased toxicity and no significant improvement in overall survival. Although angiogenesis appears to have an important role in prostate cancer, the results of anti-angiogenic therapy in castration-resistant refractory disease have hitherto been disappointing. There are various possible explanations for this lack of efficacy in castration-resistant refractory disease: redundancy of angiogenic pathways, molecular heterogeneity of the disease, loss of tumor suppressor protein phosphatase and tensin homolog (PTEN) expression as well as various VEGF-A splicing isoforms with pro- and anti-angiogenic activity. A better understanding of the molecular mechanisms of angiogenesis may help to develop effective anti-angiogenic therapy in prostate cancer.

## 1. Introduction

Prostate cancer is the most commonly diagnosed cancer in men in the Western world, with a median age at diagnosis of 66 years [1]. There will be an estimated 160,000 new cases and 30,000 deaths in 2018 in the USA, representing 19% of all new cancer diagnoses and 9% of all cancer related deaths, respectively [2]. In the United Kingdom, over 47,000 men are diagnosed with prostate cancer every year, with over 330,000 men currently living with the disease [3]. The purpose of this literature review is to assess whether angiogenesis is important in prostate cancer and, if so, whether anti-angiogenic therapies are effective in the treatment of prostate cancer. To begin with, the current treatment options in prostate cancer will be discussed, along with a summary of what is already known in relation to angiogenesis in cancer. This will be followed by the literature review on angiogenesis and anti-angiogenic therapies in prostate cancer, specifically. Finally, the discussion will consider any treatment difficulties that have emerged in such studies.

## 2. Background

### 2.1. Prostate Cancer

Prostate cancer is characterized by slow to moderate growth. Consequently, many cases are indolent and in up to 70% of incidentally diagnosed cases over 60 years death is due to an unrelated cause [4]. The five-year relative survival rate for men diagnosed in the USA between 2001 and 2007 with local or regional disease was 100%, whilst the rate for distant disease was 28.7% [5]. UK statistics show similar results: the five-year relative survival for prostate cancer was 100% in localized disease and 30% in distant disease for patients diagnosed during 2002–2006 in the former Anglia Cancer Network [6]. Most cases of prostate cancer are diagnosed by prostate specific antigen (PSA) testing or rarely by rectal examination. Prostate cancer can present with decreased urinary stream, urgency, hesitancy, nocturia, or incomplete bladder emptying, but these symptoms are non-specific and are infrequent at diagnosis [7].

### 2.2. Treatment Options in Prostate Cancer

Prostate cancer staging is divided into four stages. Stage 1 and 2 cancers are localized to the prostate whilst stage 3 cancers extend into the periprostatic tissue or the seminal vesicle, without involvement of a nearby organ or lymph node and with no distant metastasis [8]. Stage 4 tumors represent those that have spread to nearby or distant organs or lymph nodes [8].

Stage 1 tumors and stage 2 tumors of low and intermediate risk (Table 1) can be followed up by ‘watchful waiting’ or active surveillance and monitoring [9,10]. Watchful waiting has no curative intent, whilst active surveillance and monitoring defers treatment with curative intent to a time when it is needed [9]. Therefore, in active surveillance and monitoring therapy is reserved for tumor progression, with a 1–10% mortality rate [9].

Radical prostatectomy is a treatment option for localized tumors in patients with few comorbidities. Although this provides an improvement in disease progression compared to active surveillance and monitoring, it does not translate into a statistical difference in mortality: 10-year cancer-specific survival rates were 98.8% with active surveillance and monitoring compared to 99% with radical prostatectomy [9]. Complications of radical prostatectomy include the mortality and morbidity associated with major surgery and anaesthesia, penile shortening, impotence, urinary and faecal incontinence, and inguinal hernia [8].

Radiation and radiopharmaceutical treatment options include external-beam radiation therapy (EBRT), interstitial implantation of radioisotopes into the prostate and hormonal manipulation [9]. EBRT is used with curative intent in all stages of prostate cancer, with or without adjuvant hormonal therapy. Interstitial implantation of radioisotopes is used in patient with stage 1 and 2 tumors. Short term results are similar to those seen with EBRT or radical prostatectomy, but the maintenance of sexual potency is significantly higher (86–96%) when compared to radical prostatectomy or EBRT (10–40% and 40–60%, respectively) [11].

Hormonal manipulation options include surgical castration (orchidectomy) or medical castration (LH-RH antagonists) [12]. These may be used in stage 3 or 4 cancers and can be enhanced by the addition of anti-androgenic therapy and adjuvant treatment with bisphosphonates [13]. Recently approved anti-androgen agents include abiraterone acetate, an inhibitor of cytochrome P450c17, a critical enzyme in androgen synthesis and enzalutamide, a second generation androgen-receptor–signaling inhibitor [13,14,15].

Treatment options for high stage metastatic hormone-refractory prostate cancer include active cellular immunotherapy with sipuleucel-T, which has resulted in increased overall survival in metastatic castration-resistant prostate cancer, in a double-blind, placebo-controlled, multicenter phase 3 trial [16]. This lead to its approval for the treatment of asymptomatic or minimally symptomatic patients with nonvisceral metastatic castration-resistant prostate cancer in 2010. Radium-223 dichloride is used in symptomatic patients with bone metastases and no known visceral metastases [17]. Cabazitaxel, a derivative of docetaxel, is approved as a second line chemotherapy agent [18]. Further possible treatment options to prevent bone metastases include denosumab (a monoclonal antibody that inhibits osteoclast function) [19] and bone-seeking radionucleotides (strontium chloride Sr 89) [20].

Despite a widening arsenal of new treatment options, a cure is rarely achieved in stage 4 prostate cancer, although there is astriking difference in treatment response between individual patients [21]. Such outcomes emphasize the need for research into further treatment options in hormone-refractory advanced prostate cancer. One such emerging therapeutic option is inhibition of tumor-related angiogenesis. 

### 2.3. Angiogenesis in Cancer

Angiogenesis is defined as the development of new vascular vessels from pre-existing blood vessels. It has a critical role in wound healing and embryonic development and also provides collateral formation for improved organ perfusion in ischaemia [22]. It is a multi-step process triggered by an angiogenic stimulus (Figure 1). The first step of the process is the production of proteases which degrade the basement membrane. This is followed by migration and proliferation of the endothelium, resulting in the formation of a new vascular channel [23]. 

Although angiogenesis is not entirely necessary for tumor initialization (some tumors of the brain, lung, and liver can grow along pre-existing vessels) [23], once a tumor reaches a size of more than a few millimeters, formation of new blood vessels is necessary to provide an appropriate blood supply to support tumor cell viability and proliferation. Hence, angiogenesis plays an important role in tumor progression and is now recognized as one of the hallmarks of cancer [24]. 

Angiogenesis is controlled by a delicate balance between angiogenesis inducers and angiogenesis inhibitors. In a growing cancer there is a constant production of angiogenesis inducers, including vascular endothelial growth factor (VEGF)-A, basic fibroblast growth factor (bFGF, also known as FGF), angiogenin, tumor necrosis factor (TNF)-α, granulocyte colony-stimulating factor (G-CSF), platelet-derived endothelial growth factor (PDGF), placental growth factor (PGF), transforming growth factor (TGF)-α, TGF-β, interleukin-8 (IL-8), hepatocyte growth factor (HGF), and epidermal growth factor (EGF) [22]. This constant production of angiogenesis inducers results in increased activity of endothelial cells, as long as the production of anti-angiogenic factors is correspondingly reduced [25]. Among the angiogenesis activators, VEGF-A and bFGF are particularly important in tumor angiogenesis. The abundance and redundant activities of different angiogenesis inducers may explain the resistance or suboptimal effectiveness of anti-angiogenic therapies, when inhibitors acting only on a single angiogenesis activator are being used [25].

Under normal conditions, angiogenesis inducers are balanced by naturally occurring angiogenesis inhibitors, such as endostatin, angiostatin, IL-1, IL-12, interferons, metalloproteinase inhibitors, and retinoic acid [25,26]. These inhibitors can either disrupt new vessel formations or can help to remove already formed vascular channels. Shifting the balance towards angiogenesis inhibition can interfere with important physiological roles of angiogenesis, such as in embryo development, wound healing, and renal function. Interference with wound healing is a particularly important concern in cancer treatment, for example resulting in delayed post-operative healing [27]. Another example involves the inhibition of VEGF-A, resulting in vasoconstriction by means of elevated NO production, consequently elevating blood pressure [28], and increasing the risk of thrombogenesis, resulting in stroke or myocardial infarction. These factors can potentially limit the use of angiogenesis inhibition in cancer, on account of their potential side effects.

### 2.4. Angiogenesis Inhibition in Cancer

Although angiogenesis is an essential factor in tumor progression, by means of new vessel formation, this also means that angiogenesis inhibition may only result in inhibition of further tumor growth and may not actively eliminate the tumor. This, and the redundancy of the numerous angiogenesis inducers as listed above, explain why the utilization of angiogenesis inhibitors as a monotherapy has not proved to be as effective as initially expected [29]. Hence, angiogenesis inhibitor therapeutic regimes may require a combination of several anti-angiogenic strategies or may need to be complemented by other non-angiogenesis related chemotherapeutic agents in order to achieve an optimal therapeutic effect [30].

Based on the target of the therapeutic agent, angiogenesis inhibition can be divided into two main groups: direct and indirect inhibition [31]. Direct inhibitors target growing endothelial cells, whilst indirect inhibitors target the tumor cells or tumor-associated stromal cells. Small molecular fragments (e.g., arrestin, tumstatin, canstatin, endostatin, and angiostatin) are the products of proteolytic degradation of the extracellular matrix, and act as direct inhibitors by means of inhibition of the endothelial cell proliferation and migration induced by VEGF-A, bFGF, PDGF, and interleukins [32]. The direct anti-angiogenic effect of targeting integrins (cellular adhesion receptors), has also been demonstrated [32]; an integrin inhibitor—cilentigide—has been shown to inhibit tumor cell invasion [33]. Unfortunately, even though cilentigide acts both on tumor cells and endothelial cells and could be a prime example of multifactorial treatment, results of clinical trials have proved disappointing so far [34]. 

The most extensively clinically used direct anti-angiogenic strategy targets VEGF-A or its receptors. VEGF-A binds to its receptors to stimulate the proliferation of endothelial cells via the RAS–RAF–MAPK (mitogen-activated protein kinase) signalling pathway [35]. Bevacizumab is a humanised IgG1 monoclonal antibody against VEGF-A. It selectively binds to circulating VEGF-A, preventing its interaction with its receptor, VEGF-receptor 2, expressed on the surface of endothelial cells. Initial studies showed clinical improvement when bevacizumab was used in combination with chemotherapy in a number of cancers, without a marked increase in toxicity [36]. Subsequently it has been approved as part of a combination therapy in the treatment of various cancers, including metastatic lung, colorectal, and renal cell carcinoma, and as a single agent treatment in adult glioblastoma [37]. However, subsequent studies have revealed adverse effects, including gastrointestinal perforation, nephrotic syndrome, thromboembolism, surgical wound healing complications and hypertension [37,38].

In contrast, indirect angiogenesis inhibition involves an interplay between tumor or stromal cells and angiogenesis. One example involves the inhibition of epidermal growth factor receptor (EGFR), a tyrosine kinase receptor. Tumor cell expression and activation of EGFR induces interleukin production, which is demonstrated to promote intratumoral angiogenesis. Thus, blocking the expression and/or activity of EGFR can result in indirect inhibition of angiogenesis [39].

To summarize, a number of anti-angiogenesis drugs have already been approved and are currently used in cancer treatment. This prompts the question whether angiogenesis plays any role in prostate cancer progression and, if so, whether anti-angiogenic therapy would be effective in refractory castration-resistant prostate cancer, for which the current treatment options are limited. 

## 3. Results

### 3.1. Angiogenesis in Prostate Cancer

Currently there are no direct markers to assess angiogenic activity in prostate cancer, but it is reasonable to assume that vascular density is an indicator of intratumoral angiogenic activity. Microvessel density (MVD) is considered a good surrogate marker of angiogenic activity and has been demonstrated as a prognostic factor in various tumors, including breast and colon cancers as well as malignant melanoma [40]. MVD can be assessed by histological examination of the vasculature, either by assessing the most vascularised area of the tumor (‘hot spot’) or a random representative area. Preliminary data suggested that MVD is associated with higher tumor grade and stage, and worse outcome in prostate cancer [41,42]. Moreover, ultrasound imaging studies of haemodynamic indices have shown a higher peak intensity in high-grade tumors [43]. Later studies, however, have failed to confirm that MVD is an independent prognostic factor in untreated tumors, and no correlation has yet been established between MVD and effectiveness of anti-angiogenic treatment in prostate cancer [44]. Reasons for these conflicting results potentially include different counting methods, diferences in antibodies used, different population sizes, personal experience and pathological background [45]. A further limiting factor is the complex geometrical structure of the newly fromed vascular system, which is difficult to analyse on a two dimensional histological section [46]. Fractal geometry to estimate the surface dimension, computer aided automated image analysis, 3D models or magnetic resonance imaging could potentially be used to overcome these shortcomings, [46,47].

Another possible surrogate marker for tumor angiogenesis is by an assessment of the level of angiogenic regulators in the tumor. Both physiological and pathological angiogenesis is predominantly regulated by VEGF, which has various protein isoforms, each acting on their specific tyrosine kinase receptor at the cell surface [48]. Among the VEGF isoforms, VEGF-A has been extensively studied, and it has been demonstrated to play an important role in prostate cancer angiogenesis [49]. In addition, VEGF-A has been found to be overexpressed in prostate cancer and a high level of VEGF-A is associated with distant metastasis and a poorer prognosis [50,51,52]. Furthermore, in prostate cancer a high-level VEGF-A expression has been found not only in endothelial cells, but also in tumor cells [53]. 

These findings suggest that angiogenesis is important in prostate cancer, prompting subsequent clinical studies to assess whether anti-angiogenesis therapy is effective in the treatment of prostate cancer.

### 3.2. Anti-Angiogenesis Clinical Studies in Prostate Cancer

An unfiltered Pubmed search for the keywords “angiogenesis” and “prostate” revealed a steady increase in published papers between 2000 and 2013 (from 70 per year in 2000 to 213 per year in 2013) followed by a slow decline (down to 115 in 2018). This appears to reflect the fact that, despite the promising findings of initial studies, suggesting an important role of angiogenesis in prostate cancer, phase III clinical trials, mainly conducted after 2010, have proved disappointing so far. 

Since VEGF-A was demonstrated to be overexpressed in prostate cancer and associated with poor prognosis and metastasis, most anti-angiogenic clinical studies in prostate cancer have targeted VEGF-A. A randomizedphase II trial on bevacizumab involving 99 patients with hormone-sensitive prostate cancer showed improved relapse-free survival when bevacizumab was used alongside hormone-deprivation therapy (Table 2) [54]. A randomized, double-blind, placebo-controlled phase III clinical study of 1050 patients with prostate cancer showed some improvement in progression-free survival, but found no significant improvement in overall survival in metastatic, castration-resistant prostate cancer, when bevacizumab was used together with docetaxel chemotherapy and prednisone hormonal therapy [55]. Furthermore, bevacizumab resulted in increased toxicity and a greater incidence of treatment-related deaths [55]. This suggests that bevacizumab has some positive effect, especially on hormone-sensitive recurrent prostate cancer, but in hormone-resistant refractory tumors, in which the conventional treatment options are particularly prone to failure, adding bevacizumab treatment does not have any clinical benefit (Table 2).

Aflibercept (a hybrid protein composed of various domains of VEGF-receptors 1 and 2, fused to human immunoglobulin G1) also targets the VEGF-A pathway, by acting as a decoy receptor for VEGF-A. Unfortunately, similar to bevacizumab, in a phase III multicentre, randomizeddouble-blind placebo-controlled parallel group study in 1224 men with castration-resistant refractory tumors, aflibercept therapy combined with docetaxel chemotherapy and hormonal therapy did not show any improvement in overall survival [56]. 

Sunitinib and cediranib are small multireceptor molecule tyrosine kinase inhibitors, with a demonstrated activity against VEGF-receptors 1 and 2. Sunitininb is approved for the treatment of gastrointestinal stromal tumor, renal cell carcinoma and pancreatic neuroendocrine tumors. However, in a randomized, placebo-controlled, phase III trial of sunitinib therapy combined with hormonal therapy in 873 patients with refractory castration-resistant prostate cancer, there was no improvement in overall survival compared to placebo [57].

Furthermore, these anti-VEGF-A therapies have been associated with an increased rate of toxicity and adverse effects, resulting in the discontinuation of treatment (27% vs. 7%) [57]. These toxic and adverse effects included fatigue, asthenia, hand-foot syndrome, hypertension, bowel perforation, pulmonary thromboembolism, and gastrointestinal bleeding, seen in both pre-clinical and clinical studies [60,61]. In addition, treatment-related haematological problems also emerged in up to 20% of the patients, including lymphopenia, neutropenia, and anaemia [57].

Thalidomide is an immune-modulatory drug, which also has anti-angiogenic effects. Lenalidomide is a more potent analogue of thalidomide, with less prominent side effects. The mechanism of the anti-angiogenic effect of lenalidomide is not entirely elucidated, but appears to be through multiple mechanisms, including inhibition of VEGF-induced phosphatidylinositol-3,4,5-trisphosphate (PI3K)-Akt pathway signalling [62]. Lenalidomide therapy in non-metastatic prostate cancer in a phase I/II double-blinded, randomized study of 60 patients resulted in stabilization of the disease and a decline in PSA, with minimal toxicity [58]. A randomized, double-blind, placebo-controlled phase III trial in 1059 patients with castration-resistant refractory prostate cancer, however showed worse overall survival when lenalidomide was added to prednisone, hormonal, and docetaxel chemotherapy, compared to the placebo group [59]. There was also a 25% increase in adverse events, which included haematological side effects (34% vs. 20%), diarrhoea (7% vs. 2%), pulmonary embolism (6% vs. 1%), and asthenia (5% vs. 3%) [59].

To summarize, these findings suggest that anti-angiogenic therapy has no clinical benefit when added to chemotherapy or hormonal therapy in refractory, castration-resistant prostate cancer.

## 4. Discussion

Clinical trials that showed an association between high VEGF-A expression and tumor progression assessed VEGF-A protein levels by immunohistochemistry, ELISA methods, or mRNA levels by reverse-transcription-polymerase chain reaction (RT-PCR). Despite high VEGF-A expression in advanced prostate cancer using these methods, anti-angiogenic therapies targeting the VEGF-A pathway have failed to provide significant treatment benefits [63,64]. There are various possible explanations for resistance to anti-angiogenic therapy in prostate cancer. Redundancy of angiogenic pathways means that targeting a single pathway may result in upregulation of alternative pathways. For example, with long-term bevacizumab treatment, which blocks VEGF-A, there is upregulation of EGF, HGF, and PDGF [65]. Lindholm et al. demonstrated in breast cancer xenografts that targeting these pathways can be effective in anti-angiogenic therapy [66]. A combination of different anti-angiogenic therapies in prostate cancer has also showed some promising results: a phase II study of combined bevacizumab and lenalidomide therapy, added to docetaxel and prednisone chemotherapy and hormonal therapy in 63 patients with metastatic castration-resistant prostate cancer found that combined anti-angiogenic therapy can be safely administered, but further randomizedtrials are required to confirm clinical benefit [67].

Another reason for treatment resistance is due to the fact that prostate cancer is a molecularly heterogeneous disease,= and there is currently a lack of biomarkers that can help select those patients who are likely to benefit from anti-angiogenic therapy or that can assess response to anti-angiogenic treatment [48]. The genetic signature of the VEGF-A pathway or variations in VEGF-A or its receptors could be possible markers to predict therapy response, but these have as yet not been validated [68,69]. It is hoped that further stage III trials will be able to identify subgroups of patients who could benefit from anti-angiogenic treatment.

Resistance to sunitinib tyrosine-kinase-inhibitor has been shown to be associated with loss of the tumor suppressor protein phosphatase and tensin homolog (PTEN). PTEN is a gatekeeper protein that negatively regulates intracellular levels of PI3K and consequently suppresses the PI3K-Akt pathway, which normally promotes cell survival and growth [70]. Reinstating PTEN activity, by suppression of the PI3K-Akt pathway in in vitro studies, has been shown to restore sensitivity to sunitinib in cancer cells [70]. Loss of PTEN activity is considered a key event in prostate carcinogenesis, and reinstating PTEN activity in prostate cancer seems to be a promising tool in overcoming sunitinib resistance. In addition, activation of the PI3K-Akt pathway in tumors with PTEN deletion has been shown to be associated with repressed androgen signalling in prostate cancer, while suppression of the PI3K-Akt pathway was demonstrated to activate androgen receptor signalling [71,72]. In a similar way, suppression of the androgen signaling pathway resulted in activation of the PI3K-Akt pathway [71]. This suggests that there is a cross-talk between the androgen receptor and PI3K-Akt pathways, which would at least in part explain the castration-resistant phenotype observed in tumors with PTEN deletion. Since activation of the PI3-Akt pathway appears to play an important role in resistance to both sunatininb and anti-androgenic therapy, suppression of the PI3K-Akt pathway could help overcome difficulties in anti-angiogenic and anti-androgenic therapy. Recent preclinical studies on mouse models have shown that targeted inhibition of the PI3K-Akt pathway in castration-resistant prostate cancer resulted in both inhibited cancer cell proliferation and MVD [73,74]. Suboptimal results with bevacizumab treatment may also relate to the interaction between the androgen receptor (AR) signalling and angiogenic pathways. It has been long established that androgens upregulate VEGF-A expression [75], although the mechanism of this is not entirely understood [76]. Most recently, an interaction between epigenetic factors (Lysine specific demethylase 1 (LSD1), protein arginine methyltransferase 5 (PRMT5)) [77,78], zinc-finger transcription factors (specificity protein 1 (Sp1), Wilms tumor gene 1 (WT1), and early growth factor 1 (EGR1)) [76,79], different AR splice variants [80] and hypoxia mediated by the hypoxia-inducable factor 1 α (HIF-1α) [81] have emerged as potential mechanisms for androgen-dependent VEGF-A regulation. Furthermore, AR has been shown to regulate EGFR expression in prostate cancer cells. [82,83] In addition to the role of EGFR in indirect angiogenesis promotion through interleukin production, [39] it has also been demonstrated to upregulate VEGF-A directly and through induction of HIF-1α [84,85] (Figure 2).

The interaction and the importance of angiogenesis and hormonal therapy in tumor progression have initiated a clinical trial implementing dual targeting of angiogenesis and androgen signalling in hormone-sensitive tumors [54]. As discussed above, this phase II clinical trial, which combined short-course androgen deprivation therapy with bevacizumab, improved relapse free survival in recurrent, hormone-sensitive tumors. In addition, it has been demonstrated that androgen deprivation by castration, causes hypoxia in prostatic tumor cells [87,88]. Hypoxia consequently enhances the transcriptional activity of AR in prostatic tumor cells at low androgen levels, such as seen in castration-resistant prostate cancer [89]. It has been suggested that the activation of AR in hypoxic conditions is HIF-1α mediated [90], hence targeting HIF-1α could influence the AR stimulatory effect of hypoxia in castration-resistant prostate cancer. Recently, dual targeting of HIF-1α and AR pathways by HIF-1α inhibitors and enzalutamide, a second generation AR inhibitor, showed synergistic effect in castration-resistant prostate cancer cell lines, also resulting in decreased VEGF-A levels [81]. In addition, suppression of Sp1 binding to VEGF-A promoter resulted in significant reduction of VEGF-A level in castration-resistant prostate cancer cells [79]. However, a better understanding of the mechanism of the interaction between VEGF-A and AR is still needed to identify those patients who may benefit from dual targeting therapy [79,86].

Targeting VEGF-A also raises a further question: does inhibition of VEGF-A result in a pure anti-angiogenetic effect? Interestingly, it has been shown that VEGF-A has different splice isoforms and these different isoforms can show pro- or anti-angiogenic functions [91]. In the terminal exon of the VEGF-A gene, there are two alternative splice sites. Splicing at the proximal splice site results in the canonical angiogenic VEGF_165a_ isoform. Splicing at the distal splice site results in an alternative splicing isoform VEGF_165b_, which has been found to have anti-angiogenic effect by inhibiting vasodilation and reducing permeability [92,93]. The level of the anti-angiogenic VEGF_165b_ splice variant has also been found to be decreased in cancer cells, compared to normal tissue cells [93]. This means that, in cancer cells, there appears to be a shift towards the pro-angiogenic VEGF_165a_ splice variant at the expense of the anti-angiogenic VEGF_165b_ splice variant. The cause of this shift has not been entirely elucidated, but nuclear receptor-coregulator complexes have been shown to regulate splicing events, therefore aberrant recruitment of nuclear receptor-coregulator complexes to the VEGF promoter to promote VEGF_165a_ splicing has been suggested as a possible explanation [48,94]. Current anti-VEGF-A therapies lack isoform specificity, as the epitope of bevacizumab binds the N-terminal region of VEGF-A, which is present in all splice isoforms [95]. Thus, current anti-angiogenic therapies targeting VEGF-A function may result in both inhibition and promotion of tumor angiogenesis. However, the fact that the two isoforms appear to have different splice sites and post-translational regulation offers the possibility of selectively targeting specific isoforms. Serine-arginine protein kinase 1 (SRPK1), a kinase that phosphorylates SR-protein, appears to stimulate VEGF_165a_ splicing, whilst VEGF_165b_ splicing has been shown to be stimulated by Clk1/4, a dual specific protein kinase [96,97,98]. Investigation with SRPK1 knocked-down cell lines showed a shift towards the anti-angiogenic VEGF_165b_ isoform, while xenografts showed decreased tumor growth and decreased MVD in tumors [99]. In addition, specific inhibition of SRPK1 in a mouse tumor model has been shown to be associated with reduced tumor growth [100] (Figure 3).

Most current mainstream anti-angiogenic treatment therapies focus on direct angiogenesis inhibition. A further possible treatment option is indirect inhibition of angiogenesis, targeting an interplay between tumor or stromal cells and angiogenesis. The galectin family of proteins have emerged as playing an important role in this interplay, facilitating tumor progression. Galectins are β-galactoside-binding lectin proteins, which are overexpressed in various cancers and have been associated with poor prognosis and tumor progression in prostate cancer [101]. In addition to their intracellular function of promoting cell transformation and survival, galectins are also secreted into the extracellular space. Here they interact with cell surface receptors, resulting in suppression of the immune response and promotion of angiogenesis, likely by means of interaction with VEGF-receptor2 [102,103]. Rabinovich and colleagues identified that prostate cancer shows a unique galectin expression profile during cancer progression, and showed that galectin-1 is uniquely expressed at high levels in advanced prostate cancer [104]. This makes galectin-1 a potential target of angiogenesis therapy in advanced prostate cancer [105].

## 5. Materials and Methods 

The literature review was conducted by a Pubmed literature search engine using a collection of keywords with no restriction on publication date. The following word strings were used as keywords: “angiogenesis”[All Fields]] AND [“prostatic neoplasms”[MeSH Terms] OR [“prostatic”[All Fields] AND “neoplasms”[All Fields]] OR “prostatic neoplasms”[All Fields] OR [“prostate”[All Fields] AND “cancer”[All Fields]] OR “prostate cancer”[All Fields]. The search results were subsequently filtered by article type, specifically clinical trials and review articles. Abstracts were assessed for relevance with subsequent review of full text versions. Only phase II or III studies were included. Studies cited by these articles, but not included in the algorithm, were also manually scoped and were also subject of the review. 

## 6. Conclusions

The association of MVD and overexpression of VEGF-A with tumor prognosis in prostate cancer suggested that angiogenesis has an important role in prostate cancer progression. Supplementation of hormonal manipulation and chemotherapy with anti-angiogenesis therapy in hormone-sensitive prostate cancer showed some positive effect, further supporting the hypothesis that angiogenesis is an important factor in prostate cancer. Despite this, clinical trials in refractory castration-resistant prostate cancer hitherto have shown increased toxicity with no clinical benefit. A better understanding of the mechanism of angiogenesis may help to understand the failure of trials, possibly leading to targeted anti-angiogenic therapies in prostate cancer. These could include identification of specific subgroups of patients who might benefit from therapies, targeting tumor-suppressor genes that play a role in treatment resistance, or by identifying and selectively targeting splice variants of VEGF-A.

## Figures and Tables

**Figure 1 ijms-20-02676-f001:**
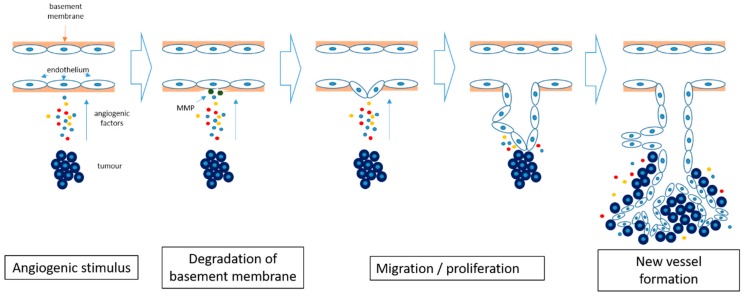
Angiogenesis in cancer. Hypoxia within the tumor induces the release of pro-angiogenic factors and results in degradation of the basement membrane by matrix metalloproteinases (MMP). The endothelial cells start to differentiate and proliferate, forming new blood vessels. The newly formed blood vessels allow further tumor growth.

**Figure 2 ijms-20-02676-f002:**
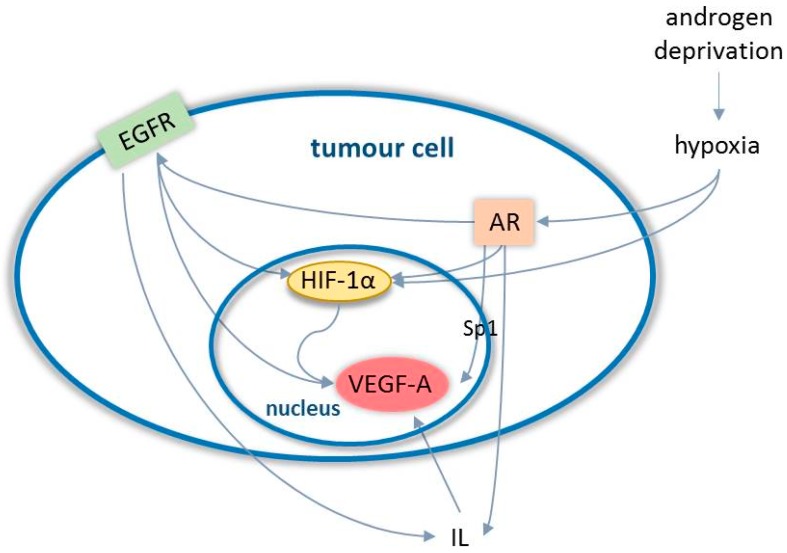
Interaction between angiogenic and androgen receptor pathways in prostate cancer cells. Castration results in androgen depletion which causes hypoxia Hypoxia enhances the transcriptional activity of androgen receptor (AR) at low androgen levels, as seen in castration-resistant prostate cancer. The activated androgen receptor promotes the overexpression of vascular endothelial growth factor A (VEGF-A) through hypoxia-inducable factor 1 α (HIF-1α) and (specificity protein 1 (Sp1) related mechanisms and also via regulation of epidermal growth factor receptor (EGFR) expression and upregulation of cytokins, mainly interleukin (IL)-6. [86].

**Figure 3 ijms-20-02676-f003:**
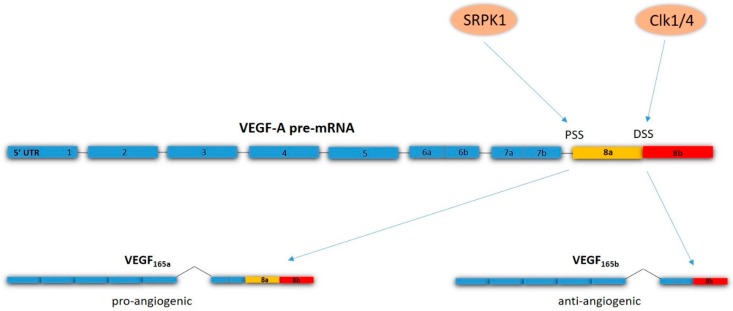
Alternative splicing of VEGF-A. Splicing at the proximal splicing site (PSS) is stimulated by serine-arginine protein kinase 1 (SRPK1), and results in the pro-angiogenic VEGF_165a_ splice variant. Clk1/4 stimulates splicing at the distal splicing site (DSS), which results in the anti-angiogenic VEGF_165b_ isoform.

**Table 1 ijms-20-02676-t001:** Risk stratification of localized prostate cancer according to NICE guidance, UK [10]. Gleason score: histological pattern of the tumor. Stage T1–T2a: tumor involving <50% of one lobe. Stage T2b: tumor involving ≥50% of one lobe. Stage T2c: tumor involving both lobes. NICE stands for the National Institute for Health and Care Excellence. PSA stands for Prostate-Specific Antigen.

Level of Risk	PSA Level (ng/mL)		Gleason Score		Clinical Stage
Low risk	<10	*and*	≤6	*and*	T1–T2a
Intermediate risk	10–20	*or*	7	*or*	T2b
High risk	>20	*or*	8–10	*or*	≥T2c

**Table 2 ijms-20-02676-t002:** Anti-angiogenesis clinical studies in treatment of prostate cancer.

Drug	Mechanism of Action	Phase of the Clinical Trial	Number of Patients	Outcome
**Bevacizumab**	Recombinant humanized monoclonal antibody that blocks VEGF-A	II	99	Improved relapse-free survival [54]
		III	1050	No improvement in overall survival [55]
**Aflibercept**	Binds to circulating VEGF-A	III	1224	No improvement in overall survival [56]
**Sunitinib**	Receptor tyrosine kinase inhibitor	III	873	No improvement in overall survival [57]
**Lenalidomide**	Multiple mechanisms, including inhibition of VEGF-induced PI3K-Akt pathway signalling	I/II	60	Disease stabilisation, decrease in PSA [58]
		III	1059	Worse overall survival [59]

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
