# Peer review of "Targeting Angiogenesis in Prostate Cancer"

_ijms, 2019, doi:10.3390/ijms20112676_

Round 1

Reviewer 1 Report

I appreciate the time and effort by authors during the revision of this manuscript. Regrettably, the revised manuscript is still below the expectations for a review article on IJMS. My previous concerns are not fully addressed, and the newly added information is not satisfactory.  

For instance, the majority of the listed clinical trials in Table 2 are for CRPC therapy while the newly added Fig. 2 is about hormone-sensitive prostate cancer. 

There are many formatting and style problems. 

For example, Table 2- Lenalidomide is cited by Ref. 60 which is a paper about Multiple Myeloma. 

Fig. 2 is poorly designed. 

 Therefore, I am afraid that I cannot recommend this manuscript to be considered for further re-evaluation. 

Author Response

see Academic Reviewer comments

Reviewer 2 Report

Al issues have been addressed. The manuscript is acceptable in its present form.

Author Response

Many thanks

This manuscript is a resubmission of an earlier submission. The following is a list of the peer review reports and author responses from that submission.

Round 1

Reviewer 1 Report

The current review article tried to list the updated advances in the application of anti-angiogenic therapy for prostate cancer. The manuscript above is an exciting and relevant topic for a review. Although the authors tried very hard, the review is far from sufficient or satisfactory.
A major concern is that the Androgen Receptor (AR) signaling pathway as the main driver of prostate cancer is entirely ignored. Considering the considerable influence of the second generation of AR inhibitors such as abiraterone acetate or enzalutamide on the survival rate of mCRPC patients, the potential impact of AR inhibition on angiogenesis should be discussed.  Also, the authors should provide some schematic diagram to describe the cross talk between the AR and angiogenesis signaling pathways. Good figures are essential to a review article.
The “introduction” section is full of unrelated and outdated information. For instance, on Line 83-86, by citing a paper published in 2011, sipuleucel-T is introduced as a potential and under investigation therapy for prostate cancer while is has been approved by the FDA in 2010.
Another major concern is that the academic standard of writing is highly variable throughout the paper, with some sections being poor indeed. The authors are highly encouraged to seek expert academic editorial assistance.
Sections entitled “Results” and “Material and Methods” in this review article seems inappropriate since this manuscript is neither a systematic review nor meta-analysis.  
Besides, there is no review on the pre-clinical studies while a significant portion of the available literature is based on in vitro and in vivo observations.  
The review would be of interest for the non-experts in the field while I cannot recommend it in its current format to be considered for publication on IJMS.

Reviewer 2 Report

In the present manuscript, Melegh and Oltean have reviewed the state of the art of the published literature on currently available strategies targeting angiogenesis to control prostate cancer. The manuscript is correctly written, easy to follow and very well focused on the subject of interest. 

Authors have reviewed most of the available updated literature. However, I suggest to the authors to include and discuss in the manuscript the works from Dr. Gabriel Rabinovich´s groups about the role of Galectins (especially Galectin-1) on angiogenesis in prostate cancer. That group has provided very interesting and cutting edge data about the pro-angiogenic and immunosuppressive role of galectins in cancer that are of major relevance to be discussed in the present manuscript.

Besides that recommendation, this reviewer considers the manuscript of high quality and suitable for publication.   

Reviewer 3 Report

This is an interesting review on the role of anti-angiogenetic drugs in prostate cancer; to this end, the authors analyzed phase II and III trials.

My comments are as follows:

- each discussion on antiangiogenic therapy focuses on a central theme: which target? As the authors correctly state, both MVD and VEGF-A have been assessed as surrogate angiogenic markers; nevertheless, there are issues in their assessment: (1) MVD may be measured by immunohistochemical antibodies against endothelial markers, by imaging techniques and even by computer-based 3D models, in order to overcome limitations due to its geometrical complexity; this topic is reviewed in Taverna et al., Front Oncol 2013;3:15. (2) According to references 43 to 45, the assessment of VEGF-A poses some problems as well: the vast majority of the literature assesses VEGF-A by immunohistochemistry (tissue samples) or by immunoassay (plasma samples). In both cases, the protein is measured on a protein level, and treatment efficacy shows no correlation with protein expression, unlike other molecular antibody-based therapeutic settings (such as HER2 or the PD-1/PD-L1 axis). The authors should discuss this points

- page 2, lines 61-64: current clinical guidelines recommend treatments options, including watchful waiting and active surveillance, on the basis of risk stratification (and not only tumor stage). Please reformulate this paragraph and cite proper references.